# Molecular Transmission Dynamics of Primary HIV Infections in Lazio Region, Years 2013–2020

**DOI:** 10.3390/v13020176

**Published:** 2021-01-25

**Authors:** Lavinia Fabeni, Gabriella Rozera, Giulia Berno, Emanuela Giombini, Caterina Gori, Nicoletta Orchi, Gabriella De Carli, Silvia Pittalis, Vincenzo Puro, Carmela Pinnetti, Annalisa Mondi, Marta Camici, Maria Maddalena Plazzi, Andrea Antinori, Maria Rosaria Capobianchi, Isabella Abbate

**Affiliations:** 1Virology and Biosafety Laboratories Unit, “Lazzaro Spallanzani”-IRCCS, National Institute for Infectious Diseases, 00149 Rome, Italy; lavinia.fabeni@inmi.it (L.F.); gabriella.rozera@inmi.it (G.R.); giulia.berno@inmi.it (G.B.); emanuela.giombini@inmi.it (E.G.); maria.capobianchi@inmi.it (M.R.C.); 2Laboratory of Clinical Pathology, San Camillo-Forlanini Hospital, 00149 Rome, Italy; caterina.gori@inmi.it; 3AIDS Referral Center, “Lazzaro Spallanzani”-IRCCS, National Institute for Infectious Diseases, 00149 Rome, Italy; nicoletta.orchi@inmi.it (N.O.); gabriella.decarli@inmi.it (G.D.C.); silvia.pittalis@inmi.it (S.P.); vincenzo.puro@inmi.it (V.P.); 4HIV/AIDS Clinical Department, “Lazzaro Spallanzani”-IRCCS, National Institute for Infectious Diseases, 00149 Rome, Italy; carmela.pinnetti@inmi.it (C.P.); annalisa.mondi@inmi.it (A.M.); marta.camici@inmi.it (M.C.); maria.plazzi@inmi.it (M.M.P.); andrea.antinori@inmi.it (A.A.)

**Keywords:** HIV primary infection, spread and epidemiology, phylogenetic analysis, ultra-deep sequencing

## Abstract

Molecular investigation of primary HIV infections (PHI) is crucial to describe current dynamics of HIV transmission. Aim of the study was to investigate HIV transmission clusters (TC) in PHI referred during the years 2013–2020 to the National Institute for Infectious Diseases in Rome (INMI), that is the Lazio regional AIDS reference centre, and factors possibly associated with inclusion in TC. These were identified by phylogenetic analysis, based on population sequencing of *pol*; a more in depth analysis was performed on TC of B subtype, using ultra-deep sequencing (UDS) of *env*. Of 270 patients diagnosed with PHI during the study period, 229 were enrolled (median follow-up 168 (IQR 96–232) weeks). Median age: 39 (IQR 32–48) years; 94.8% males, 86.5% Italians, 83.4% MSM, 56.8% carrying HIV-1 subtype B. Of them, 92.6% started early treatment within a median of 4 (IQR 2–7) days after diagnosis; median time to sustained suppression was 20 (IQR 8–32) weeks. Twenty TC (median size 3, range 2–9 individuals), including 68 patients, were identified. A diagnosis prior to 2015 was the unique factor associated with inclusion in a TC. Added value of UDS was the identification of shared quasispecies components in transmission pairs within TC.

## 1. Introduction

HIV transmission continues to occur worldwide, despite expanded access to antiretroviral therapy (ART). A considerable proportion of the new diagnoses still involves people with high-risk sexual behaviors such as men having sex with men (MSM), despite the increasing use of pre-exposure prophylaxis (PrEP) [1,2].

To design better tailored public health HIV control and prevention initiatives, a more accurate understanding of HIV transmission dynamics is needed. To this respect, molecular surveillance plays an important role. Molecular surveillance for HIV is commonly based on phylogenetic analysis of *pol* sequences, obtained in the context of antiretroviral resistance diagnosis [3,4]. Most of the molecular epidemiology studies have focused on HIV transmission among chronically infected subjects [5,6,7,8,9], but the spread of HIV-1 infection is largely driven by individuals who have recently acquired the infection, due to the high levels of virus in blood and genital secretions during this phase, referred to as PHI [10,11,12,13,14,15]. To this respect, there are few studies aimed at analyzing the transmission clusters (TC) among patients with PHI. The study was aimed at evaluating the dynamics of HIV transmission in the Lazio region during the years 2013 to early 2020, by identifying HIV TC, and factors associated with probability of patients to be included in TC. In addition, as UDS has proven to be useful for the analysis of HIV quasispecies [16,17,18,19] and has been used for the analysis of TC of HIV and of other infections [5,20,21,22], we performed a detailed analysis of *env* quasispecies harbored by individuals included in the largest TC involving B subtype.

## 2. Materials and Methods

### 2.1. Study Population, HIV Serodiagnosis, and Therapy

Study population was composed by all adult individuals with PHI diagnosis performed at National Institute for Infectious Diseases in Rome (INMI) Virology laboratory from January 2013 to June 2020 enrolled in the local established at INMI observational cohort ALPHA (approved by the INMI Ethics Committee on 13 December 2011) and SIREA (SIndrome REtrovirale Acuta, ethical committee approved on 18 February 2014).

All patients diagnosed with HIV at our Clinical Center from June 2013 were consecutively enrolled in ALPHA Study before starting cART, in order to investigate viral decay and adherence. All patients diagnosed with HIV-1 PHI were included in SIREA Study. Demographic, laboratory, clinical and therapeutic data along with blood specimens were collected and recorded for all participants in an anonymous form at baseline and, afterward, at each time point. Virological and immunological data were collected at T0, week 2, week 4, week 8, week 12, week 24, week 36, and week 48 in ALPHA Study and at baseline (defined as the date of ART initiation), day 2, week 2, week 4, week 8, week 12, week 24, week 36, and week 48 and thereafter every six months for SIREA Study. Self-reported adherence and quality of life are assessed at each time points except for day 2 in SIREA.

The inclusion criteria for our study population were: adult subjects >18 years old with PHI diagnosis based on the PHI definition that follows:(a)HIV-1 RNA levels >10,000 copies/mL and negative fourth generation HIV assay (Ab/Ag Combo), or positive fourth generation HIV assay with negative or indeterminate WB test;(b)positive HIV Ab/Ag Combo test accompanied by positive WB without p31 reactivity, associated with an antibody avidity index ≤0.80;(c)documented negative HIV-1 test performed within 6 months before the current HIV-1 diagnosis.

None of the PHI subjects of the present study was receiving PrEP at the moment of the enrolment. SIREA protocol provided early start of treatment, soon after PHI diagnosis, with 2 nucleoside reverse transcriptase inhibitors (NRTI) + boosted darunavir (DRV/b) + integrase strand transfer inhibitor (INSTI) or 2NRT I+ dolutegravir (DTG); in the subgroup of Alpha enrolled patients, who started ART within 24 weeks after PHI diagnosis, as recommended by current national guidelines first-line regimen was administered. The most recent Italian guidelines for HIV treatment are available at [23].

Laboratory HIV serodiagnosis was carried out with the fourth generation HIV-1/2 Architect HIV Ag/Ab Combo assay, Abbott (Illinois Park, IL, USA). Reactive serum samples underwent confirmation with HIV-1 Western Blot (WB, New Lav I Bio-Rad, Hercules, CA, USA), using WHO criteria (two *env* products reactivity). Avidity test [24] was carried out, on confirmed positive samples lacking p31 reactivity, to discriminate between recent (<6 months) and late infections. HIV-1 RNA was measured in plasma samples using Abbott Real-Time HIV-1 assay (Abbott Molecular, Inc., Des Plaines, IL, USA, 40 copies/mL lower limit detection).

All PHI were further stratified as follows: those in the stages from I to IV of Fiebig classification [25] were considered as “acute infections”; those in stages from V to VI were considered as “recent infections”.

Time to therapy-driven viral suppression was calculated as the day from therapy start in which a first of two consecutive not detectable HIV-1 RNA was observed, during a period of at least 24 weeks.

### 2.2. HIV-1 Drug Resistance Testing, Subtype Assignment, TC Identification, and Infection Dating by pol Population Sequencing

HIV-1 *pol* genotyping was performed on plasma samples, as previously described [26]. Transmitted drug resistance (TDR) was evaluated by considering the WHO 2009 list [27], with the additional RT mutations K65E/N, E138G/K/Q/R, V179L, G190Q, T215N, H221Y, F227C, and M230I, reported in the International AIDS Society list [28] and/or the HIVdb V.8.9-1 (https://hivdb.stanford.edu/). HIV-1 strains were defined as resistant if carrying at least one TDR mutation.

For each sequence, HIV-1 subtype was determined by molecular phylogeny. To improve the accuracy of analysis for recombinant forms, RDP3 software was used (http://web.cbio.uct.ac.za/~darren/rdp.html). Moreover, subtype classification was confirmed by the HIV Resistance database (https://hivdb.stanford.edu/), and by the COMET subtyping tool (https://comet.lih.lu/), as previously described [29].

In order to visualize recent HIV transmission in our study population a maximum likelihood (ML) tree was constructed using all the 229 *pol* sequences and TC including pairs (2 individuals) and clusters (≥3 individuals) were deduced. To avoid the influence of convergent evolution at antiretroviral drug resistance mutations, sequences were stripped at positions related to drug resistance. Only clusters with a bootstrap value ≥90% and an average genetic distance ≤0.015 substitution/site were selected.

The ML tree was inferred with the general time-reversible nucleotide substitution model (GTR) with gamma-distribution among site rate heterogeneity, a proportion of invariable sites (G + I + Γ_5_) [30] and 1000 bootstrap replicates by using MEGA 6 software [31].

The GTR + I + Γ model was considered the best one by the MEGA 6 model test, as showing the lowest BIC (Bayesian Information Criterion) score.

The tree was rooted using a midpoint rooting by FigTree software v. 1.4.2 (http://tree.bio.ed.ac.uk/software/figtree/).

Finally, the Bayesian phylogenetic tree was reconstructed with MrBayes, using a GTR + I + Γ_5_.

The Monte Carlo Markov Chain (MCMC) search was run for 5 × 10^6^ generations with the trees sampled every 100th generation (with a burn-in of 10%) [32]. Statistical support was obtained by calculating the posterior probability of each monophyletic clade, and a posterior consensus tree was generated after 10% burn-in.

Clades were considered TC only if a posterior probability of ≥0.90 was inferred.

Time of the Most Recent Common Ancestor (tMRCA) was estimated on the greatest subtype B TC that were also confirmed by UDS *env* analysis. The MCMC approach, implementing GTR + I + Γ model was used [33], as executed in BEAST software 1.81. Seventeen reference sequences of HIV-1 subtype B (see Appendix A) within the HIV-1 *pol* sequences involving in these TC, were used.

As coalescent priors, different parametric demographic models (constant population size, exponential, and logistic growth) and a non-parametric Bayesian skyline plot (BSP) were compared under strict and relaxed clock conditions (log-normal). The best combination of models was selected after testing several alternative models for each prior category, by calculating the Bayes factor (BF) with TRACER v. 1.6.

Dated trees were estimated by using a lognormal prior distribution for the clock rate, and a relaxed uncorrelated lognormal clock was finally used [33].

MCMC simulations were run for 50 × 10^6^ steps, sub-sampling parameters every 1000 steps, with a 10.0% burn-in. Convergence of parameters was assessed by calculating the Effective Sample Size (ESS) using TRACER v. 1.6 [34], after excluding an initial 10.0% for each run. All parameter estimates for each run showed ESS values >250. The trees were summarized in a target tree by the Tree Annotator program included in the BEAST package by choosing the tree with the maximum product of posterior probabilities (maximum clade credibility) after a 10.0% burn-in.

Dated trees were estimated by using a lognormal prior distribution for the clock rate, and a relaxed uncorrelated lognormal clock was used [34].

MCMC simulations were run for 50 × 10^6^ steps, sub-sampling parameters every 1000 steps, with a 10.0% burn-in. Convergence of parameters was assessed by calculating the Effective Sample Size (ESS) using TRACER version 1.6 [35], after excluding an initial 10.0% for each run. All parameter estimates for each run showed ESS values >250. The trees were summarized in a target tree by the Tree Annotator program included in the BEAST package by choosing the tree with the maximum product of posterior probabilities (maximum clade credibility) after a 10.0% burn-in. Phylogeny was analyzed visually in FigTree v. 1.4.2.

### 2.3. UDS of HIV Env Region

Plasma HIV-1 RNA extraction was performed for all patients infected with B subtype included in TC by *pol* population sequencing, for whom sufficient stored plasma samples were available, by using QIAamp Viral RNA kit (Qiagen, Hilden, Germany). *Env* amplification was performed by a nested PCR. A first one-step RT-PCR was carried out with Platinum quality proofreading reverse transcriptase (Invitrogen, by Life Technologies, Monza, Italy): outer sense primer ATGGGATCAAAGCCTAAAGCCATGTG (position 6556–6581 in HXB2), outer antisense primer AGTGCTTCCTGCTGCTCCCAAGAACCCAAG (position 7822–7792 in HXB2). Retrotranscription was performed at 55 °C for 30 min; then first round PCR was carried out by 30 cycles (94 °C for 2 min, 94 °C for 30 s, annealing at 60 °C for 30 s, extension at 68 °C for 1 min and 30 s, and a final elongation at 68 °C for 5 min). To maximize the genetic heterogeneity to be amplified and sequenced, replicate PCR were pooled, representing the content of 1 mL of plasma.

The second PCR composed by 30 cycles (94 °C for 2 min, 94 °C for 30 s, annealing at 60 °C for 30 s, extension at 68 °C for 30 s, and final elongation at 68 °C for 5 min) was performed using the proofreading DNA polymerase Platinum^®^ Taq DNA Polymerase High Fidelity (Invitrogen, by Life Technologies): inner sense primer TACAATGTACACATGGAATT (position 6958–6977 in HXB2), inner antisense primer ATTACAGTAGAAAAATTCCCC (position 7382–7362 in HXB2).

Amplicon library was generated using Ion plus Fragment Library kit, according to the manufacturer’s instructions (Thermo Fisher Scientific, Waltham, MA, USA). The library was barcoded with Ion Xpress™ Barcode Adapter and ligated with the P1 adaptors. Sequencing was performed on Ion S5 sequencer (Thermo Fisher Scientific). The reads were corrected with an in-house developed pipeline that excluded all reads with a length <400nt and/or a mean quality score of <20. Pollux algorithm (https://doi.org/10.1038/s41598-017-08139-y) was used to specifically correct insertions and deletions in homopolymeric regions.

Reads were clustered with the CD-HIT software (100% identity cut-off). All clusters that were not represented by at least one forward and one reverse sequence, or with less than five reads were excluded. To evaluate the accuracy of the correction, a plasmid with *env* amplicon was sequenced in parallel by UDS and Sanger method. All differences found in UDS with respect to Sanger sequences were considered errors. At the end of the correction pipeline the overall error rate was 0.000087 ± 0.00017 (mean ± STD).

To perform the ultra-deep phylogenetic analysis, the ML tree was inferred using the GTR model with gamma-distribution among site rate heterogeneity, a proportion of invariable sites (GTR + I + Γ_5_), and 1000 bootstrap replicates (Mega 6). To calculate diversity, a number of 5000 sequences were randomly sorted for each patient. Pairwise genetic distance analysis was conducted using the Tajima–Nei model [36].

HIV-1 coreceptor usage was inferred from each patient consensus sequence by using the Geno2Pheno (G2P) algorithm (https://coreceptor.geno2pheno.org/) [37], setting false-positive rate (FPR) at 10%. Sequences with FPRs of ≥10.0% were considered R5 tropic.

### 2.4. Statistical Analysis

Univariable logistic regression analysis was used to identify the factors associated with presence in molecular TC as the outcome, and age, gender, subtype, year of diagnosis and other factors, as indicated in Table 1, as possible explanatory variables.

*p* values < 0.05 were considered statistically significant. All analyses were performed using SPSS V.23 for Windows.

## 3. Results

### 3.1. PHI Description

Among the 270 PHI diagnosed in our Virology Laboratory, during the study years, only 229 PHI were from subjects enrolled and followed in INMI observational cohorts, for which samples and clinical data were collected. Clinical, demographical and risk behavior features of the study subjects are shown in Table 1. Briefly, the enrolled subjects showed a median age of 39 (IQR 32–48) years; 94.8% were males; 86.5% Italians; 83.4% MSM. An increasing trend of MSM proportion (2014–2019: 12.6%–21.4%, *p* = 0.05) was observed over time in the overall study population. Considering the sub-classification as “acute” and “recent” infections, 51.1% of cases were in the first group, characterized by a higher HIV-1 RNA viremia (median log_10_ copies/mL: 6.15; IQR 5.4–7.0 versus 5.0; IQR 4.2–5.5, *p* < 0.001). The proportion of “acute” vs “recent” infections did not significantly increase over time (2013–2019: 48.3%–64.2%, *p* = 0.95).

HIV-1 B subtype was predominant (56.8%), followed by F1 (9.2%), CRF02_AG (7.9%), C (6.6%), A1 (5.2%), and CRF12_BF (3.9%). An increase of non-B subtypes proportion, although not significant, was found over time (2013 vs. 2019: 12.1% vs. 21.2%, *p* = 0.74); the increase remained not significant also when considering MSM (2013 vs. 2019: 10.8% vs. 25.7%, *p* = 0.35).

Overall, TDR prevalence was 10.0%, among which TDR to non-nucleotide RT inhibitors (NNRTI) was predominant (5.2%). Prevalence of TDR to other drug classes was 1.3% for nucleos(t)ide RT inhibitors (NRTI), and 3.9% for protease inhibitors (PI).

Of 229 enrolled individuals, 212 (92.6%) started treatment soon after diagnosis (early ART), within a median of 4 (IQR 2–7) days; for 190 patients who started early ART, sustained virological suppression was reached in a median time of 20 weeks (IQR 8–32); for 22 of those who received early ART, the follow-up period was insufficient to establish the achievement of sustained virological response.

### 3.2. Identification of TC and Study of the Factors Associated with the Inclusion in a TC

The phylogenetic tree constructed with 229 *pol* sequences, deriving from all the subjects enrolled in the study, identified 20 TC (Figure 1). In these 20 TC, sequences from 68 persons enrolled in the study were included (30%). TC cluster size ranged from 2 to 9 subjects. The same results were obtained both with ML and Bayesian analyses. As shown in Figure 1, 8 TC involved 2 subjects, and 12 clusters involved ≥3 patients. Individuals in TC were mostly males (97.1%) with a median (IQR) age of 37 (32–44), Italians (92.6%), MSM (86.8%) and infected with B subtype (58.8%). In 51.5% of cases, an acute infection (from Fiebig II to IV) was observed. Regarding viro-immunological parameters, individuals in TC had a median viral load of 5.6 (IQR 4.6–6.5) log_10_ copies/mL, and a median CD4 cell count of 548 (IQR 370–743) cell/mm^3^. Only in one TC, including 4 individuals (5.9%) infected with B subtype, a virus strain carrying a protease inhibitor resistant mutation (I85V) was found (Figure 1). Among the analyzed factors, no significant differences were found between subjects included in small (size 2) and bigger (≥3) TC.

By comparing the characteristics of patients included versus those not included in TC, the only factor that was found significantly associated with TC inclusion was a year of diagnosis prior to 2015 (years 2013–2014 vs. 2015–2016, *p* = 0.03). In 2013–2014, the proportion of patients in TC vs. out of TC, was 45.6% vs. 28.6%, thereafter, along the subsequent years of the study, the differences in frequencies among people in TC vs. out TC became thinned. Having reached early control of virus replication was not associated with TC exclusion/inclusion status. The same result was obtained even when considering as “non sustained suppressed patients” those who did not start ART within 24 weeks from diagnosis (Table 1).

### 3.3. Ultra-Deep Analysis of TC

In order to more accurately analyse infection transmission chains, we performed UDS of the high variable *env* V3 loop coding region. The analysis was focused on subtype B infections, that, in fact, represent the most common single subtype represented in our study population (Figure 1). All residual samples available from subjects included in TC based on *pol* population sequencing (*n* = 31) underwent UDS, that was successful (sufficient coverage) for 25 patients. A median of 18,605 (IQR 11,321–26,494) of sequences/Pt were obtained. Overall, median diversity was 0.005 (IQR 0.003–0.008) substitutions per site. Regarding viral tropism, all patients carried predominant R5 viruses, with a median percentage of false positive rate of 38 (IQR 25.3–65.2).

The phylogenetic tree constructed with ultra-deep sequences highlighted eight TC: five TC were pairs (Pt 14 and Pt20; Pt15 and Pt45; Pt10 and Pt11; Pt26 and Pt17; Pt12 and Pt19) and three clusters (A, B and C) included ≥3 subjects. Cluster A was composed by Pt2, Pt9, Pt8, Pt3, Pt1, Pt4, Pt5, and Pt7; cluster B by Pt21, Pt22, Pt23, and Pt27; cluster C by Pt31, Pt30, and Pt32 (Figure 2, panel A). Overall, the composition of the 8 TC found by *env* UDS was superimposable to that obtained by *pol* population sequencing.

We further focused on the three largest TC in UDS (A, B, and C). All of them originated around the year 2009; (95% Highest Posterior Density (HDP) 2005–2012 for cluster A; 2005–2011 for cluster B; and 2006–2012 for cluster C), as resulted by molecular clock Bayesian analysis.

Cluster A: this cluster included two sub-clusters, (A1 and A2). Subcluster A1 involved Pt 5 and Pt7, showing overlapping quasispecies with intermixed sequences; subcluster A2 included three patients, of whom two (Pt1 and Pt4) showed overlapping quasispecies with intermixed sequences, and one patient (Pt3), with completely segregated quasispecies. Other patients of cluster A (Pt2, Pt8 and Pt9), showing completely segregated HIV quasispecies, did not form subclusters.

Cluster B included 1 subcluster (B1), composed by two patients with overlapping quasispecies with intermixed sequences (Pt 21 and Pt22), and two patients (Pt23 and Pt27) with segregated quasispecies.

Cluster C did not show subclusters, and included three patients (Pt30, Pt31, and Pt32), with completely segregated quasispecies.

It is to be underlined that pairs of patients showing overlapping quasispecies shared close time of serodiagnosis, and were characterized by an overlapping duration (Figure 2B) of high HIV viral load (not shown), consistent with being members of a donor/receiver couple. This hypothesis could be actually confirmed for Pt5/Pt7, and for Pt21/Pt22, based on available behavioral information collected at diagnosis interview.

## 4. Discussion

Molecular surveillance of circulating viruses is a powerful tool to identify and monitor epidemic spread [3,4,5,6]. Several studies, including those from our group, have been focused on the molecular analysis of HIV from chronically infected patients [5,6,7,8,9]. However, molecular surveillance applied to PHI infections are expected to provide a more realistic snapshot of the epidemic dynamics, allowing the analysis of the viral genome close to the time of transmission. To this respect, few studies have been focused on recently acquired infections.

Our study analyzed 229 among the 270 PHI diagnosed at INMI during the period January 2013–June 2020 (from a total of 1390 new HIV diagnoses). These represented more than 50% of all PHI taken in care in our Italian region (Lazio) and for this reason provided a reliable description of the landscape of PHI in our territory [38].

To our knowledge, our study involved the largest number of patients in Western countries, and the results are in keeping with those from a previous study performed on patients from Thailand [39]. In the present study PHI involved predominantly Italian males (94.8%) and MSM (83.4%), suggesting that MSM population has a greater chance of transmitting HIV between their members and in contributing in active clusters, in line with several studies [6,7,8,9,40].

About 30.0% of individuals from our cohort were involved in TC, as observed in other studies including both acute/recent infections [41] and newly diagnosed individuals [8,9,42,43]. However, the proportion of patients belonging to TC observed in our study was higher as compared to other studies on TC from USA, Asia, and Europe [8,9,44], and to a study from the Thailand PHI cohort (14.1%) [39], where the only factor found to be associated with TC inclusion was the younger age.

In the present study cluster size was smaller (from 2 to 9 individuals) than that observed in other studies focused on newly diagnosed but not PHI infections [7,8], but was similar to that from the Thailand PHI cohort [39]. It is to be noted that, since we analyzed only our study population with 229 sequences, using a threshold of genetic distance <0.015 for clustering, only recent TC were identified. This may represent a limit of our study. In fact, higher distance thresholds would have yielded larger clusters. On the other hand, in this case it would be more probable that individuals falling in the same cluster were related only by indirect transmission events, as pointed out in [45].

Regarding risk factors for HIV transmission, most PHI patients included in TC were MSM (86.8%), as in the overall PHI population.

Considering virological factors, the stage (i.e., early vs. recent) of PHI as well as viral load at the diagnosis, HIV subtype and TDR did not seem to influence the inclusion in TC. In line with previous studies, the overall TDR prevalence was about 10% [8,39,46,47,48,49]. The predominant role in the overall spread of TDR was played by NNRTI mutations, that were observed in 5.9% of patients included in TC, similar to that found in the overall population (5.2%). Only one TC, including patients with B subtype, showed all components carrying the surveillance PI resistance mutation I85V [27].

In addition, in the majority of our patients, early therapy was administrated soon after PHI diagnosis, and this could have restricted the infection spread and consequently the size of the respective TC.

However, the inclusion TC was not found related to time to viral suppression during early therapy. Moreover, the subgroup of patients who were not treated during PHI and, consequently, were viremic for a long time, was not found to be more likely associated to TC.

In our study neither younger age, nor gender, nor nationality, nor risk factor for HIV transmission, nor TDR were found significantly associated with inclusion in TC, except a year of diagnosis prior to 2015. It is reasonable that the early initiation of ART, which was implemented around 2015, and whose clinical benefit was definitively demonstrated by the results of the INSIGHT and Temprano studies [50,51], could have indirectly exerted a reduction of HIV transmission during PHI, resulting in a decreased risk of early treated people to be included in a TC after this time.

A peculiar aspect of the present study is that UDS was applied to obtain deeper insights into cluster composition, based on *env* sequencing. The first result of this analysis was the confirmation of the TC composition obtained by *pol* population sequencing. This was not implicit, since in other studies the phylogenetic trees obtained by different HIV regions were not superimposable [52]. In addition, UDS provided the opportunity to study the quasi-species harbored by each patient and to explore whether intermixed populations could be observed, to identify direct chains of transmission among individuals. In fact, three couples of subjects with intermixed viral quasi-specie were observed, for two of which a direct connection could be ascertained. Although it is clear that UDS is not indispensable to conduct molecular epidemiology analysis, from the present results it is evident that it represents an advancement in phylogenetic analysis, helping in reconstructing epidemiological links between individuals when these are unknown.

In conclusion, this study represents a deep analysis of HIV-1 spread in a local area and may provide new instruments to better understand the infection dynamics in the context of acute infections.

## Figures and Tables

**Figure 1 viruses-13-00176-f001:**
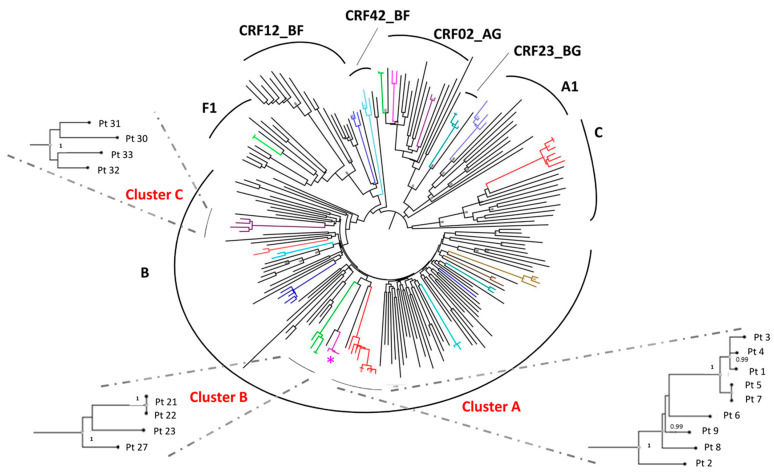
**Phylogenetic tree constructed with *pol* population sequences according to subtypes.** Bayesian phylogenetic tree constructed by all *pol* sequences from the study subjects (*n* = 229) highlighted 20 colored TC (8 pairs and 12 TC ≥3 individuals) using a genetic distance ≤0.015. The asterisk shows the resistant cluster carrying the protease inhibitor resistance I85V. In the inserts, the largest TC of subtype B are shown, with the significance of the sub-cluster nodes (posterior probability). Abbreviation: Pt, patient.

**Figure 2 viruses-13-00176-f002:**
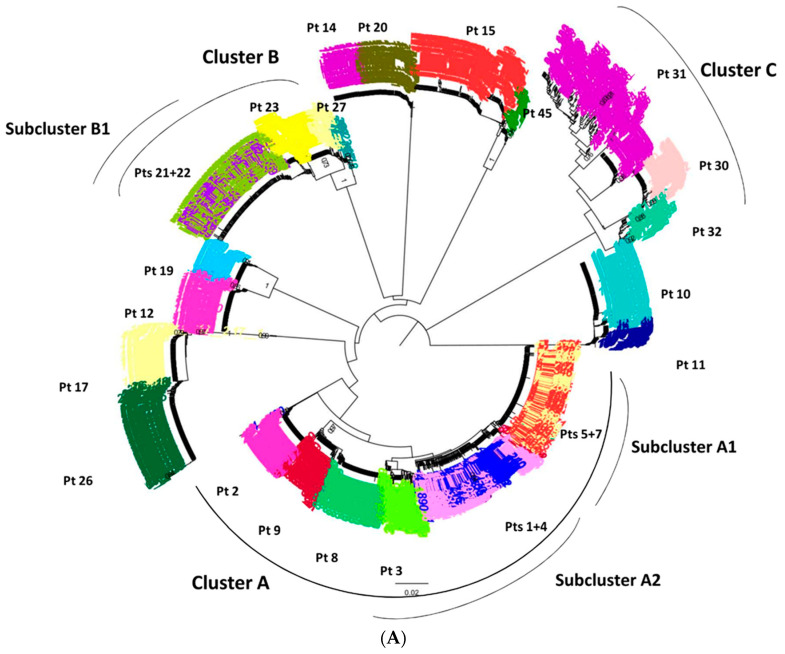
Phylogenetic tree constructed with ultra-deep *env* region sequences of subtype B infected subjects, previously included in TC by *pol* population sequencing with viremic period (**A**) Maximum likelihood phylogenetic tree constructed with all the representative *env* sequences obtained from subtype B subjects included in TC, corresponding to *pol* population sequences shown in Figure 1. (**B**) Viremic period of each patient included in the clusters A, B, and C starting from the diagnosis to a time in which all the included subjects displayed a detectable viremia. Each patient is represented with a different color and Pt numbers are the same of Figure 1.

**Table 1 viruses-13-00176-t001:** Primary HIV infection (PHI) patient’s characteristics and factors associated with HIV-1 TC.

Characteristic	Overall	In Cluster	Out of Cluster	Univariate Analysis	*p* Value ^a^
	*n* = 229	*n* = 68 (29.7%)	*n* = 161 (70.3%)	OR (95% CI)	
**Gender (male vs. female **^†^**), *n* (%)**	217 (94.8)	66 (97.1)	151 (93.8)	0.46 (0.09–2.15)	0.32
**Age (years), median (IQR)**	39 (32–48)	37 (32–44)	39 (32–49)	0.98 (0.96–1.00)	0.18
**Year of diagnosis, *n* (%)**					
2013–2014	77 (33.6)	31 (45.6)	46 (28.6)	**2.21 (1.10–4.44)**	**0.03**
2015–2016 **^†^**	77 (33.6)	18 (26.5)	59 (36.6)	1	
2017–2018	61 (26.6)	16 (23.5)	45 (28.0)	1.17 (0.54–2.54)	0.70
2019–2020	14 (6.1)	3 (4.4)	11 (6.8)	0.89 (0.23–3.56)	0.87
**Nationality, *n* (%)**					
Italians **^†^**	198 (86.5)	63 (92.6)	135 (83.9)	1	
Foreigners	31 (13.5)	5 (7.4)	26 (16.1)	2.43 (0.89–6.61)	0.08
**Risk factor, *n* (%)**					
MSM	191 (83.4)	59 (86.8)	132 (82.0)	1.44 (0.64–3.23)	0.38
Heterosexual **^†^**	35 (15.3)	8 (11.8)	27 (16.8)	1	
IV drug addict	3 (1.3)	1 (1.5)	2 (1.2)	1.19 (0.11–13.3)	0.89
**Fiebig stage, *n* (%)**					
II-IV **^†^**	117 (51.1)	35 (51.5)	82 (50.9)	1	
V-VI	112 (48.9)	33 (48.5)	79 (49.1)	1.02 (0.58–1.80)	0.94
**CD4 cell count (cell/mm^3^), median (IQR)**	557 (389–712)	548 (370–743)	557 (391–705)	1.00 (0.99–1.00)	0.98
**Viral load (log_10_ copies/mL), median (IQR)**	5.5 (4.6–6.4)	5.6 (4.6–6.5)	5.4 (4.6–6.3)	1.04 (0.83–1.30)	0.74
**Subtype ^b^, *n* (%)**					
B **^†^**	130 (56.8)	40 (58.8)	90 (55.9)	1	
Non B	99 (43.2)	28 (41.2)	71 (44.1)	1.13 (0.63–2.00)	0.68
**Transmitted drug resistance, *n* (%)**					
Any drug class	23 (10.0)	4 (5.9)	19 (11.8)	0.47 (0.15–1.43)	0.18
**Time from diagnosis to virological suppression, *n* (%) of patients undergoing early ART ^c^**					
Overall	190 (83.0)	57 (83.8)	133 (82.6)		
≤20 weeks **^†^**	100 (52.6)	29 (50.9)	71 (53.4)	1	
>20 weeks	90 (47.4)	28 (49.1)	62 (46.6)	0.90 (0.49–1.68)	0.75
**Patients who did not start ART within 24 weeks from diagnosis, *n* (%)**	17 (8.2)	7 (10.4)	10 (7.0)	1.63 (0.59–4.50)	0.34

^a^ Boldface indicates factors that were significantly associated (*p* < 0.05) with transmission clusters. ^b^ 11 B clusters (2–9 individuals) and 9 non-B clusters (2–6 individuals). ^c^ Patients who started early ART, for whom sufficient follow-up observation period was available. **^†^** Reference group (dummy). Abbreviations: MSM, Men who have Sex with Men.

## Data Availability

HIV *pol* sequences are available in GenBank (accession numbers: MW489589- MW489817).

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
