# Peer review of "Molecular Transmission Dynamics of Primary HIV Infections in Lazio Region, Years 2013–2020"

_viruses, 2021, doi:10.3390/v13020176_

Round 1

Reviewer 1 Report

Fabeni et al aimed to identify the HIV transmission clusters in Lazio region and to investigate the factors associated with the inclusion in transmission clusters, using molecular methods. Their analysis revealed that of 229 patients enrolled in the study, most of them were Italians, MSM infected with subtype B. Twenty clusters with a range of 2-9 individuals, including 68 patients, were identified. In addition, only the less recent year of diagnosis (2013-2014) was identified as a factor associated with the inclusion in a cluster.

See my comments/suggestions below.

Title

Line 2. The term “dynamics” is misleading. I would suggest being replaced with “transmission dynamics”.

Materials and methods

Line 100. Apart from the phylogenetic analysis, was any further analysis for recombination performed?

Line 102. Usually, this kind of analysis is performed by using a high number of globally sampled sequences as references and separately for each subtype. Authors analysed only their sequences (n=229) and considered as TCs those including the most genetically similar sequences (d<0.015). By this way you may identify only the most recent TCs and not all of them. This (i.e. that only the recent TCs were identified) should by stated and clarified all over the text. In addition, this has also to be mentioned at the discussion as a limitation of the study.

Lines 106-128. A justification should be added for all the parameters used in these analyses (i.e. models, runs, burn-in etc). For instance, why the GTR+I+Γ was used? Did the authors perform a test to find the most suitable model for their data? And if so, which was this test and what was the probability of the selected model versus the others? Furthermore, there are updated versions available for many programs that were used (i.e. Mega X, Beast 1.84, BEAST 2 etc).

Lines 112, 123 and 127. The percentage of the burn-in was set at 50%. Most studies use a much lower percentage for the burn-in, usually equally to 10%. Why it was set at 50% in these analyses? In addition, why it was set at 50% for the analysis performed by TreeAnnotator and MrBayes and at 10% for the analysis performed by BEAST? Did authors test for convergence in their MCMC results?

Lines 115-116. Previously in the text it was mentioned that only clusters with a bootstrap value >90% and an average genetic distance <0.015 substitution/site were selected (lines 104-105). Here it is stated that clades with a posterior probability of >0.90 were considered as epidemiological clusters. Epidemiological clusters are different from the molecular transmission ones? I suggest not, but in order to avoid any misunderstanding I advise authors to use one term for the clusters all over the text and also to summarize at the end of this section all the criteria used to specify the clusters as TCs.

Lines 117-120. The time of the Most Recent Common Ancestor (tMRCA) was estimated only for the subtype B TCs. It has to be justified why only these TCs were selected. Which were the criteria for this selection? Based on Figure 1, I suppose that it was not the size of the subtype B TCs since there are TCs with the same size in other subtypes (i.e. subtype C).

Lines 120-121. Since the sampling coverage of the study is high (>50%), BDM (Birth Death Models) are more appropriate for the molecular clock analysis compared to Coalescent, which is suggested when the sampling coverage is very low. Additionally, more details have to be given for the distributions and the models selected for this analysis and also if informative priors were used.

Line 128. The dated tree (maximum clade credibility tree) is usually interpreted in a non midpoint rooted form.

Lines 157-161. See my comment on lines 106-128.

Lines 166-168. Mann-Whitney and chi2/Fisher’s exact tests are usually used to test for the existence of differences among the groups, but statistical models are the most appropriate way to identify the factors associated with an outcome. I suggest authors to perform a logistic regression analysis using the presence in molecular transmission clusters as the binary outcome variable and other variables (i.e. age, gender, subtype, year of diagnosis etc) as possible explanatory variables.

Results

Lines 180-181. You cannot refer to a “nearly significant increase” with a p-value equal to 0.05. A non-significant increasing trend was found for the MSM.

Line 186. The increasing trend in MSM is mentioned twice (here and in lines 180-181) with different percentages, p-values and time periods. If the second statement refers to a sub-group of the MSM (i.e. those infected with subtype B), this has to be clarified accordingly.

Line 195. Table 1. Year of diagnosis, risk factor and subtype variables should have only one p-value estimated by the chi2/Fisher’s exact test comparing all the levels of these variables (as for nationality and Fiebig stage).

Lines 202-203. This should be rephrased since the total number of sequences identified within clusters was 68.

Lines 212-215. See my comment above (lines 166-168).

Lines 214-215. Different percentages are stated in Table 1.

Figures 1 and 2A. Please improve the resolution of the figures. In addition, you may consider for the purpose of clarity to remove the sequences’ labels and use different colours for the branches of the tree. 

Discussion

The discussion should be extended by adding the study limitations (see my comment above on line 102).

References

I suggest authors to update their references. There are recent studies on this topic that are not mentioned either in discussion or at the references.

Reviewer 2 Report

Major comments:

The manuscript needs an English edition.

Abstract

The abstract is not informative. The author needs to choose the results which support the author’s conclusion in the abstract. The aim and conclusion are not clear in the abstract.

Introduction

The first part of the sentence “Most of the molecular epidemiology studies…(p2, lines 46-47)” is contradicted to the fact that the author presented references 31-34 as the example of recent molecular epidemiology studies.

Materials and methods

The study population, HIV serodiagnosis, and therapy paragraph

  1. The study population, HIV serodiagnosis, and therapy section are not logically presented. If this study is a part of SIREA and Alpha study, the author needs to introduce these studies as well as the inclusion criteria for them then move to the PHI diagnosis.
  2. Please add a reference for the Italian national guidelines for HIV treatment if available. This is helpful as most of the readers in the world are not familiar with the treatment protocols in Italy.
  3. Please describe the follow-up protocols for these cohorts.
  4. Please refer to the latest drug resistance list from the International AIDS society in 2019.
  5. Please provide the GenBank accession numbers for the 17 reference sequences.
  6. Please register at least the pol sequences to the GenBank and add the accession numbers to the manuscript.

HIV env region ultra-deep sequencing (UDS) paragraph

  1. It is unclear how the data set from 25 out of n=31 was selected. These numbers should be moved to results and explained.
  2. Please add the condition for the first one-step RT-PCR.
  3. Please add a reference for this protocol if available.
  4. For the second PCR, the primer sets would be called “inner” not outer.
  5. Please add a reference for HIV-1 coreceptor usage, if available.

Results

3.1 PHI description paragraph

  1. The sentence “Actually the number of PHI… (p4, lines 172 to 175)” should move to the discussion part with reference of country data that support this finding.
  2. The reason why only 229 among the 238 who were classified as PHI were analyzed in this manuscript was not clear. If it was because 9 were lost in follow-up (238 minuses 229) please add the reason or describe it clearly.
  3. The time trend results are also important in this manuscript. Please summarize results (MSM proportion, the proportion of “acute” or “recent”, subtypes distribution, TDR prevalence, etc.) in an independent table.

3.2 Identification of transmission clusters (TC) and study of the factors associated with the inclusion in a TC section

  1. What does the author mean “during PHI (p6, line 217)”?

3.3 Ultra-deep analysis of transmission clusters paragraph

  1. How were the 25 individuals among 68 in the cluster selected for UDS analysis?
  2. It is not easy to understand which patients account for each cluster in figure 2A. It would become clearer if the details are written in the text for example 5 TC were paired (pt17 and pt26; pt 12 and 19 …) and 3 were composed by more than 3 individuals (Cluster A: pt 1 to 5 and 7 to 9; Cluster B) or in the figure.

Table 1

  1. The percentage of out of the cluster in 2013-2014 is 28.6% in table 1 but 30.9% in the text (p6, line 214). Which is correct?

Figure 2B

From the text, it is unclear how the author could identify the duration of the viremic phase in Figure 2B. It is also not clear at which time of the viremic phase the blood was collected from each patient for UDS analysis. The reviewer understands that all the individuals start ART either soon after PHI diagnosis in SIREA arm or within 24 weeks in Alpha arm. However, the pt 27 was viremic for nearly 3 years. When was the entry period of this patient to the cohort? A clear description of the flowchart of these cohorts and follow-up is mandatory either in the text or as a figure.

Discussion

  1. It is important to describe the proportion of Prep Use in this study population.
  2. The prevalence of TDR in the overall population was 10.0% in table 1 but 5.2% in the text (p9, line 291). Which is correct?
  3. The detailed description of the INSIGHT and TEMPRANO studies, in which years these studies were conducted, and how they influenced the size of TC are mandatory if the author thinks these studies contribute to restrict the size of TC.
  4. From this discussion, the early HIV diagnosis and immediate ART initiation strategy seem to be important to control HIV transmission. This could be done without conducting a molecular epidemiological analysis. The reviewer recommends reconstructing the discussion section (and introduction section if necessary) to discuss the meaning to conduct UDS.

Minor comments:

  1. Please spell out acronyms in the first instance in the text.
  2. References must be numbered in order of appearance in the text.
  3. Please check the reference styles once again.

Round 2

Reviewer 1 Report

Statistical analysis and Table 1.

The reference category is missing and should be added in all the categorical variables which were used as possible explanatory variables in the logistic regression analysis (Table 1). Furthermore, since the sum of the percentages of the subgroups in many of the categorical variables (i.e. year of diagnosis, risk factor, Fiebig stage, subtype) is 100%, I cannot understand how the reported p-values were estimated in these variables since none of the subgroups was set as the reference category for the analysis. For instance, according to Table 1, those diagnosed during 2013-2014 have a 2 times higher probability to have been infected within a TC compared to those diagnosed at which time period?

Author Response

Rome, Jan 19 2021

Manuscript ID: viruses-1043062

Title: Molecular dynamics of Primary HIV Infections in Lazio region, years 2013-2020

Authors: Lavinia Fabeni, Gabriella Rozera, Giulia Berno, Emanuela Giombini, Caterina Gori, Nicoletta Orchi, Gabriella De Carli, Silvia Pittalis, Vincenzo Puro, Carmela Pinnetti, Annalisa Mondi, Marta Camici, Maria Maddalena Plazzi, Andrea Antinori, Maria Rosaria Capobianchi, Isabella Abbate

Reply to specific points of referee 1:

Statistical analysis and Table 1.

The reference category is missing and should be added in all the categorical variables which were used as possible explanatory variables in the logistic regression analysis (Table 1). Furthermore, since the sum of the percentages of the subgroups in many of the categorical variables (i.e. year of diagnosis, risk factor, Fiebig stage, subtype) is 100%, I cannot understand how the reported p-values were estimated in these variables since none of the subgroups was set as the reference category for the analysis. For instance, according to Table 1, those diagnosed during 2013-2014 have a 2 times higher probability to have been infected within a TC compared to those diagnosed at which time period?

Reply. We agree with the suggestion, and we recalculated for each explanatory variable in the logistic regression analysis p-values related to a defined reference group (dummy) now indicated with the symbol † in Table 1.

We replaced old Table 1 with the new data and we reported and discussed the new data obtained by this recalculation in the abstract (lines 31-32) and in text at lines 253-256 and 357-359 (highlighted in light blue in the revised version ).

With kind regards,

Isabella Abbate

Laboratory of Virology

National Institute for Infectious Diseases

"L. Spallanzani" IRCCS

Reviewer 2 Report

The updated version is well revised.

Author Response

Rome, Jan 19 2021

Manuscript ID: viruses-1043062

Title: Molecular dynamics of Primary HIV Infections in Lazio region, years 2013-2020

Authors: Lavinia Fabeni, Gabriella Rozera, Giulia Berno, Emanuela Giombini, Caterina Gori, Nicoletta Orchi, Gabriella De Carli, Silvia Pittalis, Vincenzo Puro, Carmela Pinnetti, Annalisa Mondi, Marta Camici, Maria Maddalena Plazzi, Andrea Antinori, Maria Rosaria Capobianchi, Isabella Abbate

Reply to specific points of referee 2:

The updated version is well revised.

Thank you for your revision.

With kind regards,

Isabella Abbate

Laboratory of Virology

National Institute for Infectious Diseases

"L. Spallanzani" IRCCS

This manuscript is a resubmission of an earlier submission. The following is a list of the peer review reports and author responses from that submission.